# The Brave New World of Early Treatment of Multiple Sclerosis: Using the Molecular Biomarkers CXCL13 and Neurofilament Light to Optimize Immunotherapy

**DOI:** 10.3390/biomedicines10092099

**Published:** 2022-08-28

**Authors:** Andrew Pachner

**Affiliations:** Department of Neurology, Geisel School of Medicine at Dartmouth and Dartmouth-Hitchcock Medical Center, Lebanon, NH 03756, USA; andrew.r.pachner@dartmouth.edu

**Keywords:** multiple sclerosis, inflammation, demyelination, CNS injury, molecular biomarkers, immunotherapy

## Abstract

Multiple sclerosis (MS) is a highly heterogeneous disease involving a combination of inflammation, demyelination, and CNS injury. It is the leading cause of non-traumatic neurological disability in younger people. There is no cure, but treatments in the form of immunomodulatory drugs (IMDs) are available. Experience over the last 30 years has shown that IMDs, also sometimes called disease-modifying therapies, are effective in downregulating neuroinflammatory activity. However, there are a number of negatives in IMD therapy, including potential for significant side-effects and adverse events, uncertainty about long-term benefits regarding disability outcomes, and very high and increasing financial costs. The two dozen currently available FDA-approved IMDs also are heterogeneous with respect to efficacy and safety, especially long-term safety, and determining an IMD treatment strategy is therefore challenging for the clinician. Decisions about optimal therapy have been particularly difficult in early MS, at the time of the initial clinical demyelinating event (ICDE), at a time when early, aggressive treatment would best be initiated on patients destined to have a highly inflammatory course. However, given the fact that the majority of ICDE patients have a more benign course, aggressive immunosuppression, with its attendant risks, should not be administered to this group, and should only be reserved for patients with a more neuroinflammatory course, a decision that can only be made in retrospect, months to years after the ICDE. This quandary of moderate vs. aggressive therapy facing clinicians would best be resolved by the use of biomarkers that are predictive of future neuroinflammation. Unfortunately, biomarkers, especially molecular biomarkers, have not thus far been particularly useful in assisting clinicians in predicting the likelihood of future neuroinflammation, and thus guiding therapy. However, the last decade has seen the emergence of two highly promising molecular biomarkers to guide therapy in early MS: the CXCL13 index and neurofilament light. This paper will review the immunological and neuroscientific underpinnings of these biomarkers and the data supporting their use in early MS and will propose how they will likely be used to maximize benefit and minimize risk of IMDs in MS patients.

## 1. Introduction

Multiple sclerosis (MS) is a highly heterogeneous disease involving a combination of inflammation, demyelination, and CNS injury. It is the leading cause of non-traumatic neurological disability in younger people. Most persons with MS (PwMS) develop the relapsing-remitting form, relapsing-remitting MS (RRMS), which initially presents as an initial clinical demyelinating event (ICDE), also called clinically isolated syndrome (CIS), usually in the form of optic neuritis or myelitis (in this review, the term ICDE will be used instead of CIS to avoid confusion, since some clinicians currently use the term CIS to apply only to those patients who after evaluation of ICDE do not meet McDonald 2017 criteria for RRMS [1]). Persons with ICDE (PwICDE) as a group are highly heterogeneous with regard to future clinical activity [2,3,4,5], and in approximately one quarter of PwICDE [6], the event is monophasic and further clinical or MRI events do not occur. Even for many PwICDE who at their initial event satisfy the latest, highly sensitive criteria for MS [1], they may have little or no MS activity after the first event, as defined by new attacks, new demyelinating lesions on MRI, or disability progression [7]. In a landmark study of long-term outcomes, Chung et al., found that 30 years after the initial event, 33% remained defined as CIS and had not developed MS [8].

There is no cure for MS, but treatments in the form of immunotherapy using immunomodulatory drugs(IMDs) which dampen neuroinflammation are available. Experience over the last 30 years has shown that IMDs, also called disease-modifying therapies (DMTs), are effective in downregulating neuroinflammatory activity, decreasing the frequency of MS attacks, and improving the prognosis of patients with highly inflammatory forms of MS. However, there are a number of negatives in IMD therapy, including potential for significant side-effects and adverse events [9], uncertainty about long term benefits re disability outcomes [10,11], and very high and increasing financial costs [12,13]. The currently available FDA-approved IMDs also are highly heterogeneous with respect to efficacy and safety, especially long term safety, and determining an IMD treatment strategy is therefore extremely challenging for the MS clinician [14].

Prediction of future course of MS early in the disease has long been a goal of neurologists. Confavreux et al., in 2003 [15] identified a number of clinical variables that were helpful, including gender, age, symptoms and course (relapsing-remitting vs. progressive) at onset of the disease, degree of recovery from the first relapse, time to a second neurological episode, and the number of relapses in the first 5 years of the disease. Molecular biomarkers grounded in the biology of the disease would be especially helpful, and many biomarkers have been proposed. Biomarkers can be classified in a number of ways, including how they would be used in patient care [16]. Examples are: CSF OCBs(diagnosis), neutralizing antibodies to interferon-beta or natalizumab(loss of therapeutic efficacy) [17,18], absolute lymphocyte count(biomarker for adherence to sphingosine-1-phosphate receptor blockers). A thorough review of all potential biomarkers for MS is beyond the scope of this review; a number of extensive reviews of this area have been published [16,19,20,21]. Currently, the only molecular biomarker in broad clinical use in MS is CSF oligoclonal bands(OCBs) representing immunoglobulin production within the CNS. Excessive immunoglobulin in the CSF was described first in MS in the 1940s [22]. This 80 year old biomarker has proven helpful enough for aiding in the diagnosis of MS to be included in diagnostic criteria for MS [1,23]. However, the predictive value of OCBs for future MS activity in PwICDE is poor, especially because of false-positives [24], i.e., many OCB+ PwICDE do not develop future activity. The use of this biomarker in the diagnosis of MS has improved diagnosis of MS by decreasing the time to diagnosis of RRMS after ICDE, but at considerable cost, i.e., diagnosing many patients with MS who ultimately do not have the disease [25,26]. Thus, the presence of CSF OCBs is a helpful **diagnostic** biomarker, but is not predictive enough to be used as a **predictive** biomarker to guide therapy in the broad spectrum of PwICDE.

In a complex disease such as MS, there are multiple potential uses of biomarkers, e.g., aiding in the prediction of the development of MS in at-risk individuals such as first degree relatives of PwMS, diagnosis of MS, prediction of disease inflammatory activity, identification of ongoing CNS injury, or response to treatment. Here, we will focus on two uses of biomarkers in MS: prediction of future neuroinflammatory activity and correlation with neuronal damage.

## 2. Two Logical Candidates for Molecular Biomarkers Based on the Biology of MS

The pathology of MS includes neuroinflammation, neuronal injury, and demyelination. Promising biomarkers are currently being studied for the first two of these processes, while no molecular biomarkers for the demyelination of MS are on the immediate horizon. A host of molecular biomarkers in the CSF and blood have been proposed, and a complete review of all of these is beyond the scope of this review, but two molecular biomarkers stand out as particularly promising: the CXCL13 index and serum neurofilament light.

### 2.1. Intrathecal Production of CXCL13 as a Neuroinflammation Predictive Biomarker

#### 2.1.1. Need for a Neuroinflammation Predictive Biomarker

IMDs, all of which are designed to suppress neuroinflammation, are currently the mainstay of MS therapy, but choosing the best therapy in patients with MS(PwMS) is extremely challenging. This is especially true early in the disease. In PwICDE, IMDs would ideally be initiated as early as possible in patients likely to develop inflammatory forms of MS. In PwICDE as a group, IMDs delay the time to developing clinically definite MS(CDMS) [27,28,29,30], but assessing benefit/risk ratios without reliable predictive biomarkers is nearly impossible, given the how benign the course in some PwICDE can be. Imaging of the CNS is somewhat helpful, but suffers from a number of problems, including the “clinical-radiological paradox” [31]. Thus, ideally relatively benign subpopulations of PwICDE, which can only currently be identified in retrospect after years of followup, would be identified at the time of the ICDE by the absence of a single biomarker or combination of biomarkers predictive of MS activity, and IMD treatment for this benign subpopulation might be limited to no or low-risk IMDs. This would be particularly beneficial in pandemic times, since “high-efficacy” but higher risk IMDs interfere with immune responses to vaccines [32]. Conversely, treatment with “high efficacy” IMDs (HE IMDs) [33,34,35,36], with a higher risk profile, might be better suited for patients with positive neuroinflammation predictive biomarkers.

#### 2.1.2. The Biological Basis for CXCL13 as a Prognostic Biomarker

A logical candidate for a neuroinflammation prognostic biomarker is a molecule or molecules involved in recruitment of B cells and key T cell subsets into the CNS. CXCL13 is a small (10.3 kDa) chemokine, first described as B-lymphocyte chemoattractant in 1998 [37] and shown to be a key molecule in the process of lymphoid follicle formation in secondary lymphoid organs(SLOs). It is a major chemoattractant for B cells and some T cell subsets, including follicular T helper cells(TFH) [38] and likely other T cell populations that help B cells [39]. CXCL13 is the only ligand binding to the receptor CXCR5, found on B cells and TFH. B cells are highly involved in MS pathogenesis [40], and their depletion, using B cell depletion therapies(BCDT) such as the anti-CD20 monoclonal antibodies rituximab, ocrelizumab, and ofatumumab, reduces the formation of new inflammatory lesions and relapses in relapsing-remitting MS (RRMS) patients, making BCDTs one of the most effective therapies in MS [41,42]. Another key cell type implicated in MS is TFH [43,44], which both promotes B cell activity and shares the surface receptor CXCR5 with B cells. The role of B cells and TFH cells contribute to MS pathogenesis is not yet well-defined and is a topic of active research [45,46,47].

CXCL13 has also been implicated in the development of ectopic lymphoid follicles in the CNS in MS. Whereas the secondary lymphoid organs(SLOs), the lymph nodes and spleen, are sites of maintenance of mature naïve lymphocytes, lymphocyte activation by antigens, clonal expansion, and affinity maturation, lymphocytes can also reside in ectopic lymphoid follicles(ELFs) in chronically inflamed tissues, such as in joints in rheumatoid arthritis, thymus in myasthenia gravis, rejecting transplants, and some cancers. These sites have also been called tertiary lymphoid organs or structures(TLOs or TLSs), ectopic lymphoid structures(ELSs), or ectopic germinal centers(EGCs). Although infiltration of immune cells into non-lymphoid tissue sites, including the CNS, can be a random and diffuse process, ELFs are locally inducible within chronically inflamed sites and can contain many of the organizational, cellular, and molecular features found in lymph nodes and spleen. However, the extent to which ELFs mimic the structure of SLOs is variable, from poorly organized lymphocyte aggregates to highly organized structures producing almost all of the cytokines and chemokines in SLOs. One of the best studies of CNS ELFs in MS or its models was in the Theiler’s murine encephalomyelitis virus(TMEV) model of MS [48]. Expression of chemokines and cytokines related to lymphocyte accumulation, trafficking and SLO organization such as CXCL12, CXCL13, CCL19, IL-17, and LTa, as well as local antibody production, was prominent in lymphoid aggregates. However, other features of SLO were absent in TMEV such as discretely organized T and B cell follicles, supportive FDC stromal cells, Ki-67+ proliferating and caspase-3+ cells indicating ongoing antigen-driven selection, and germinal center GL7hi B cells. This partial replication of ELFs is common in other diseases and their models [49,50,51,52,53].

The role of ELFs in the pathogenesis of MS is controversial. Prineas was the first to identify “reticular-like cells embedded within lymphoid-like structures and lymphatic capillaries within old plaques” [54], and since then ELFs have been described in pathological specimens from MS patients by a number of investigators [55,56,57]. In one of the original articles describing ELFs in patients with secondary progressive MS (SPMS) [55], the investigators found “a network of follicular dendritic cells producing CXCL13”. A recent review article on ELFs in MS [58] includes “presence of stromal/follicular dendritic cells expressing CXCL13” as one of the defining features of CNS ELFs in MS. However, ELFs have been generally only present in patients with progressive forms of MS, primarily those with SPMS. Additionally, the percentage of SPMS patients in whom ELFs are not found is 50% [59]. Thus, the importance of ELFs to the biology of MS and the role of CXCL13 in the establishment and maintenance of ELFs, is unclear at this point in time.

CXCL13 can also be found strongly upregulated in the inflamed CNS in MS models [48,60,61,62]. In MS models, or in biopsied or autopsied MS brain, the intrathecal production of CXCL13 can be measured directly in CNS tissue by measuring CXCL13 mRNA or identifying CXCL13 by immunostaining [63]. However, in PwMS, tissue is not available, and most studies have measured CXCL13 in the available body fluids, CSF and/or serum.

At this point it needs to be stressed that the key biological process to be measured for the use of CXCL13 as a biomarker in MS is **intrathecal** production of CXCL13, i.e., its production within the CNS. Since CNS tissue in PwMS is not available in routine practice to allow direct measurement of CXCL13 mRNA or protein, intrathecal production can be measured indirectly by measuring the CXCL13 index, i.e., correcting CSF concentrations for serum concentrations and the integrity of the blood-CSF barrier, as described in section #4 below.

#### 2.1.3. Methods for Measuring CXCL13

There has been an increasing interest in CXCL13 as a biomarker in MS. A PubMed search of “CXCL13”,”biomarker”,and “multiple sclerosis” reveals that prior to 2008 there were only 3 articles identified while after 2008 there were 72 such papers.

Most studies in the literature have utilized enzyme-linked immunoassays(ELISAs) to measure CXCL13. Its clinical use in neurology has been up until now primarily to measure CXCL13 in the CSF in the diagnosis of Lyme neuroborreliosis(LNB) [64,65,66,67,68,69,70], the infection of the CNS with the spirochete *Borrelia burgdorferi* [71]. Although ELISA measurement of CXCL13 has a relatively poor lower limit of quantitation(LLOQ), with the LLOQ in most ELISA kits being 7.8 pg/mL up to 25 pg/mL, that does not interfere with its use in LNB, since levels of CXCL13 in the CSF in LNB tend to be very high.

Since the inflammation in MS is limited to the CNS, most of these studies have concentrated on the CSF, and despite its problem with sensitivity ELISA has been the predominant methodology used [66,72,73,74,75,76]. However, in our lab and in others [77], ELISA has proven to have inadequate sensitivity. Thus, ELISA is not an optimal method of detection of CSF CXCL13 in MS. Other methodologies for assaying CXCL13 in CSF and serum have been used, including immunoPCR [78], and SIMOA [79], but these are not readily available for most clinical laboratories. However, an accurate and reliable methodology for assaying CXCL13 in the CSF is Luminex (https://www.luminexcorp.com/, accessed on 13 July 2022), a microsphere-based technology with increasing use in the clinical laboratory. Two publications in the literature have used commercially available Luminex for CSF CXCL13, either using multiplex [80], or singleplex approach [25]; these studies are described more below.

#### 2.1.4. Studies of CXCL13 as a Biomarker in MS

After demonstration of CXCL13 in the brains and CSFs of MS patients in 2004 [55,81], Krumbholz et al. [63] published in 2006 an extensive study of CSF CXCL13 in the CSF of MS patients and found that 17/30 RRMS patients had elevated levels, while only 2 of 8 and 4 of 14 patients with primary progressive and secondary progressive MS had elevated levels. A large percentage of samples were below the limit of detection of the ELISA used in the study, i.e., 10 pg/mL. Similar data were obtained in subsequent studies [66,72,74,75,76,77,82]. A particularly interesting use of the assay was in predicting conversion to MS in patients with clinically isolated syndromes [73,83]. All of the above studies used ELISA methodology, which is flawed by its low sensitivity as noted above.

Another problem with the studies noted above is that they measured CSF CXCL13 alone without making any correction for serum levels. Since CXCL13 is a small molecule, i.e.,kDa of 10, which crosses the blood-CSF barrier much more readily than large molecules, serum CXCL13 contributes greatly to the CSF CXCL13 concentration, and absence of correction for serum levels is a weakness of the above studies. This is especially true because of the wide variability in serum CXCL13 levels in PwMS, from 16.3 to 524 pg/mL in our experience. It is also important to correct for blood-CSF barrier integrity since the albumin quotient in MS patients is highly variable. The issue of “a potential effect of pathological transfer of CXCL12 or CXCL13 from blood to CSF” was addressed by Krumbholz by separately analyzing MS patients with a “disrupted blood–brain barrier” defined in their work by a ratio of CSF to serum albumin(i.e., Q_alb_) of greater than 0.0074. However, none of the above studies used Reiber’s formulae for transfer of proteins from blood into CSF [84]. Alvarez et al., in 2015 published the first such study assessing the CXCL13 index (I_CXCL13_), i.e., (conc CXCL13_CSF_/conc CXCL_serum_//conc albumin_CSF_/conc albumin_serum_) which corrected for serum concentrations as well as integrity of the blood–brain barrier [85]. This study, however, was not testing the index’s predictive ability for neuroinflammation, but rather how I_CXCL13_ could be used to predict optimal response to B cell depletion in PwMS.

Intrathecal production of CXCL13 is not unique to the neuroinflammation of MS, and can be found in other neuroinflammatory diseases. It is especially strongly upregulated in Lyme neuroborreliosis [86,87]. Thus, the utility of the CXCL13 index as a **diagnostic** biomarker for demyelinating disease is limited, both for false-positives as noted above, and false negatives. I.e., a significant proportion of PwICDE who have a low risk of significant future neuroinflammation have normal CXCL13 indices [24], and thus cannot be differentiated in the initial diagnostic evaluation from patients with non-inflammatory neurological disease

#### 2.1.5. Using the I_CXCL13_ to Predict Neuroinflammation in Early MS

As of the late spring of 2022, there are 21 FDA-approved treatments as “disease-modifying” in MS [88], and all are IMDs, designed to target the immune response, i.e., to downregulate the neuroinflammation of MS. Some are mildly immunomodulatory having “moderate” efficacy (ME-IMD) and lower risk, while others are immunosuppressive with “high” efficacy (HE-IMD), but higher risk. Choosing among these IMDs is extremely challenging for the clinician, and the literature does not provide significant guidance. There has been a trend toward initiating HE-IMD early in the disease [33,34,35,36]. However, early MS patients are young people, and the risks for decades of treatment with immunosuppressive drugs are considerable and difficult to evaluate [89], requiring large pharmacovigilance studies [90]. Even a short (2 year) mean duration of treatment of MS patients with B cell depletion led to a 3.8 fold increase in serious infections requiring hospitalization compared to the general population [91]. HE-IMDs also suppress responses to vaccines [32], a particularly disturbing problem in the COVID era [92].

Given this challenging situation, a readily available biomarker early in the disease course that would be predictive of future neuroinflammation would be very helpful. CSFs are commonly obtained in PwICDE, especially since 2017 when new criteria for the diagnosis of MS included the use of CSF OCBs in fulfilling criteria for dissemination in time in PwICDE. In a study of 67 PwMS and 67 patients with non-inflammatory neurologic disease [24], CSFs and sera were obtained as part of a biobank program in the Department of Neurology at Dartmouth-Hitchcock Medical Center. A total of 41 of the 67 MS spectrum patients had ICDE, a very early opportunity to initiate IMDs. The ICDE patients were followed for a mean of 2.6 years during which time neuroinflammatory activity was determined as the presence of clinical relapses or new gadolinium-enhancing lesions, or new or unequivocally enlarging T2 lesions [93]. In this study, the I_CXCL13_ was found to be an excellent biomarker for the prediction of future disease activity. Because of the range of CXCL13 concentrations in blood, there is a similar range of CXCL13 concentrations in CSF in patients with non-inflammatory neurological disease, but the I_CXCL13_ values in those patients are low, usually less than 15.

Especially exciting in the above study was its utility in patients with the ICDE where the negative predictive value(NPV) was 91%, i.e., the absence of an elevated I_CXCL13_ was highly predictive of the absence of future inflammatory activity in this group of patients. In contrast, the NPV for OCBs was only 64%. The index was less predictive in its positive predictive value of 53%, i.e., the situation where a positive index predicted the presence of future activity; this number was still higher than the 29% found for the PPV of OCBs. These data raise the possibility that the I_CXCL13_ could be used at the time of ICDE to determine therapy; i.e., PwICDE with negative I_CXCL13_ could be treated with either no IMD or low-risk ME-IMDs, while PwICDE with elevated I_CXCL13_ would be treated with HE-IMDs. This would accomplish the objective of “personalized medicine” approach [94,95], in which treatment could be guided by predictive biomarkers.

#### 2.1.6. The Brave New World of Using the CXCL13 Index to Select Optimal Therapy—So, What Precisely Is This Vision of a Brave New World, and How Do We Achieve That Vision?

PwICDE-Since the absence of an elevated CXCL13 index in PwICDE is highly predictive of the absence of future neuroinflammation and IMDs target neuroinflammation, using HE-IMD in PwICDE without an elevated I_CXCL13_ would be contraindicated since there is substantial risk with little benefit, and these patients could be treated with either ME-IMDs or clinically followed without IMDs, depending on their clinical presentation. In contrast, a number of studies have shown that early HE-IMDs can result in better outcomes compared to the escalation approach [96,97,98,99], so patients with ICDE with an elevated I_CXCL13_ would be treated with HE-IMDs.

Acceptance of this strategy by clinicians would likely require a clinical trial [100] that demonstrates the utility of the I_CXCL13_ in early MS, especially since neurologists treating MS generally do not use HE-IMDs until they have tried ME-IMDs, and frequently not even then. Fernandez et al. [101] surveyed the treatment practices of 233 neurologists caring for PwMS in 11 European countries. For clinically isolated syndrome they would “never use” HE-IMD even if there were multiple-gad enhancing lesions as well as spinal cord lesions. Interestingly, large percentages of the respondents would not use HE-IMDs even later in the disease when the neuroinflammatory pattern was more established and protracted. Although the survey is now a bit outdated and also did not include American neurologists, it is indicative of the hesitation of many neurologists to use HE-IMDs because of their immunosuppressive properties and risks of adverse effects. The clinical trial would also need to show that the CXCL13 index outperforms MRI and OCBs, which appears likely since the latter biomarkers are unreliable for prediction of future MS activity [102].

The trend toward more aggressive therapy of early MS is exemplified by the establishment of the CELLO study(NCT04877457) in 2022. This study was designed to investigate the treatment effect of the B cell depleter ocrelizumab compared with placebo on clinical and radiological outcomes in patients with radiologically isolated syndrome(RIS), i.e., asymptomatic demyelinating appearing CNS lesions on MRI. RIS is an even earlier diagnosis than ICDE and represents a risk of developing MS. The management of RIS is highly controversial with some clinicians using IMDs while the majority do not; in a recent study only 16% of patients were treated for RIS with IMDs [103]. Interestingly, in that study IMD exposure in RIS subjects was not associated with the risk of developing a first clinical event.

The clinical trial testing the CXCL13 index in PwICDE would be complex. The optimal comparison arm for index-positive and -negative patients is not obvious, and would need to be worked out. Index-negative patients could be treated with a ME-IMD or followed without an IMD clinically and with imaging. The primary outcome measure might be either evidence of disease activity(EDA) or conversion to clinically definite MS. Repeat lumbar punctures could be performed in the various arms of the study to determine whether treatment lowered the index. A combination of biomarkers, such as CXCL13 index, NFl, and MRI, may be better than any of them alone [104].

If the clinical trial demonstrates clinical utility of the CXCL13 index, it will accomplish two goals:it will allow neurologists to identify patients with a high likelihood of future neuroinflammation who can then be treated as early as possible with HE-IMDsit will allow neurologists to identify patients with a low likelihood of future neuroinflammation who can thus be either treated with lower-risk medications or managed by clinical and imaging followup without IMD treatment.
b.Patients with RRMS- The CXCL13 index also might be helpful in some patients with RRMS. For instance, there is substantial controversy about when in the course of RRMS, IMDs can be discontinued. The course of RRMS is highly variable, but neuroinflammatory activity tends to decrease as PwMS age. Thus, older patients who have not had clinical or MRI activity for years could potentially safely discontinue IMDs; this is being investigated in a clinical trial (DISCOMS, NCT03073603). Another clinical trial might demonstrate that IMDs can be safely discontinued in older RRMS patients with normal CXCL13 indices, and this might identify patients better than other identifiers such as age.c.Patients with progressive forms of MS- IMDs are generally not indicated in patients with progressive forms of MS with one exception: progressive patients who have evidence of neuroinflammation. Siponimod, a sphingosine-1-phosphate receptor blocker was shown to be effective at delaying the progression in secondary progressive MS patients, 47% of whom had active inflammation as demonstrated by relapses within the previous 2 years or by imaging, and the drug is indicated now for SPMS patients with activity [105]. Similar results were obtained in a clinical trial of ocrelizumab in primary progressive MS(PPMS) [42] in which PPMS patients with activity had a slight but statistically significant improvement in disability progression. The CXCL13 index, which is elevated in some progressive MS patients [24], might outperform other measures in predicting improved disability outcomes with IMD treatment in progressive forms of MS.d.As a biomarker for response to treatment with B cell depletion- CXCL13 is critical for B cell recruitment into the CNS, and its intrathecal production could be a predictor of a positive response to B cell depletion with rituximab, ocrelizumab, or ofatumumab. This was the conclusion of an interesting study [85] in which 30 relapsing multiple sclerosis patients with breakthrough disease while on beta-interferon or glatiramer acetate were treated with the B-cell deplete rituximab. Optimal responders were defined as having no evidence of disease activity, and this group had a higher baseline CXCL13 index than those who developed disease activity in followup.

### 2.2. Neuronal Injury Biomarkers-Neurofilament (Nfl)

#### 2.2.1. Need for a Marker of Axonal Injury in MS

The pathology of MS consists of three major processes: inflammation, demyelination, and damage to the neuron, and particularly its axonal component. Since the only pharmaceuticals we have to modify the disease course in MS are immunomodulatory drugs(IMDs), much of the emphasis in research has been directed at the inflammatory process. Demyelination is readily identified by MRIs. However, neuroaxonal damage, the major cause of disability, has been very difficult to quantitate. Having a biomarker that could correlate with neuroaxonal damage for individual MS patients would help clinicians manage this unpredictable, chronic, disabling disease. This exciting possibility has prompted an outpouring of recent research in this area. In total, 466 manuscripts with the search words of “multiple sclerosis”, “neurofilament”, and “biomarker” are entered into the PubMed.gov database, as of July 2022; 2/3 of these papers were published in the past 3.5 years, almost all dealing with neurofilament light. Having a blood biomarker for neuroaxonal damage also has relevance to other CNS diseases such as dementias [106], stroke [107], and amyotrophic lateral sclerosis [108].

#### 2.2.2. The Biological Basis of Neurofilaments as a Measure of Neuroaxonal Damage

Unfortunately, little is known about the precise mechanisms by which neurons are injured in MS. Injury may be independent of inflammatory processes [109]. The pathology of progressive disability in the later stages of MS may be significantly different than in the earlier more inflammatory phases of the disease [110]. However, a constant in MS is damage to the neuron. Since most investigators feel that the primary site of neuronal injury in MS is to the axon, this damage will be called “neuroaxonal” in this review.

Neurofilaments (NF) are cytoskeletal protein and the structural scaffold of neurons, axons and dendrites and are composed of light (NFl), medium (NFm) and heavy (NFh) chain subunits. These are large molecules with the smallest being NFl with a molecular weight of about 70 kDa. Due to their abundance and specificity for neurons, their increased presence in body fluids represents a marker of neuronal injury. All pathological processes that cause neuroaxonal damage release NF proteins into the extracellular space and into the CSF. Neurofilament measurement, most commonly NFl, provides a quantification of the intensity of ongoing neuroaxonal damage using fourth-generation single molecule array(SIMOA) methodology [111]. Neuroaxonal injury is, of course, not specific for MS, and occurs in a wide variety of neurology diseases including dementias, Parkinson’s disease, cerebrovascular disease, traumatic brain injury, and others, and NF measurement is being evaluated in those diseases [112,113,114] as well as in MS.

It is generally assumed that NFl enters the CSF as a consequence of neuroaxonal cell membrane disruption. Once in the CSF, NFl could enter the blood via recently described CNS lymphatic pathways [115]. Although there is a significant correlation between CSF and blood NFl levels, blood levels may also be affected by cardiovascular risk factors, damage to the peripheral nervous system [116], integrity of the blood–brain barrier, increased blood volume, and impaired renal function [117]. In addition levels of Nfl in the serum in normals increase physiologically with age and decrease with body-mass index (BMI) [118].

#### 2.2.3. Methods of Measurement

NFl is readily measured in the CSF where concentrations are substantial in normal individuals, in the 500–1000 picograms/mL range, likely representing normal turnover of neurons and their proteins. Since NFl is a neuronal protein and not produced outside of the nervous system, serum levels in young control subjects are very low, about 3 pg/mL [119]. This situation is completely reversed compared to that for CXCL13 described above, where the concentration of CXCL13 in the CSF in normals is extremely low, because of almost total lack of intrathecal production of CXCL13 in the absence of neuroinflammation, while the serum concentration is relatively high because CXCL13 is produced at high levels in secondary lymphoid tissue [120]. In fact, a large percentage of CSF CXCL13 in normals derives from passive movement of the chemokine from the blood. Thus, NFl represents a unique type of biomarker very localized to the nervous system, and since MS pathology generally does not include damage to peripheral nerves, the source of NFl in MS is limited to the CNS.

#### 2.2.4. Studies of Nfl in MS

Because of the variability of serum NFl outlined above, this biomarker has been primarily utilized in group level comparisons. However, the clear advantage of accessing blood relative to cerebrospinal fluid makes blood NFl an attractive possibility for a clinically useful biomarker if some of the challenges in its use in the clinic can be overcome.

**Early studies**—One of the first descriptions of NFl in MS was by Lycke et al., who used an ELISA to measure NFl in CSF in MS patients and found that NFl concentrations were quite high and correlated with MS activity [121]; in this paper, the authors used a relatively low sensitivity ELISA, only able to detect concentrations above 125 pg/mL. Subsequent work revealed that CSF neurofilament concentrations were increased in all stages of MS even in the absence of measurable inflammatory activity [122,123]. Using an electrochemiluminescence assay, with better sensitivity than the ELISA, Kuhle and colleagues [124] confirmed the elevation of serum NFl in MS patients, and noted a correlation with white matter lesion volume. Subsequent cross-sectional studies also noted that an elevations in sNfL are associated with relapse rate, recent or impending clinical relapses, and MRI lesions [125,126,127].**sNFl measurement to measure response to treatment**—Because of the easy accessibility of blood, blood biomarkers can potentially be used to serially monitor patients. Serial sampling in large groups has been performed extensively [128,129,130,131] demonstrating consistently that in cross-sectional studies, treatment with IMDs lowers serum levels of Nfl.**An endpoint in clinical trials**—A major problem in any clinical trial of new agents in MS is the large number of patients that need to be enrolled, even in phase 2 studies. Sormani et al., have recently proposed sNFl as a potential primary endpoint in phase 2 studies that would provide proof-of-concept for larger and expensive phase 3 studies [132]. In that manuscript, the investigators calculated that for a phase 2 clinical trial with sNFl measured at month 6 as the primary endpoint(90% power, 5% significance level), if the treatment effect was a 40% reduction in the experimental arm vs. the control arm, only 28 subjects would be required per arm. This represents a significant improvement compared to the numbers needed using current outcome measures, and could potentially result in more MS medications being tested in phase 2, and then phase 3, studies.

#### 2.2.5. The Brave New World of Using NFl in Clinical Practice—So, What Precisely Is This Vision of a Brave New World, and How Do We Achieve That Vision?

Studies of NFl have been almost universally cross-sectional, and thus the optimal use of this biomarker in any single individual is not clear from these studies. There are a number of factors aside from disease activity that limit the interpretation of sNfL measurements at single time points, such as obesity, age, and blood volume and thus, the use of this biomarker may be sub-optimal due to lack of intra-individual reproduction [127,133]. Additionally, elevated sNFl may persist for long periods after a single episode of CNS injury; e.g., a recent study of several blood protein biomarkers in traumatic brain injury [134] found that elevated sNfl persisted for long periods after the damage, even for a full year, indicating a lack of sensitivity to change over time.

A recently published study of NFl concentrations in 10,133 blood samples, however, has provided extensive normative data [119], with accompanying age and BMI information, which will advance its potential as a biomarker in individual MS patients and even as a clinical trial endpoint. In an interesting study from the Czech Republic [135], sNfL levels were measured at the time of treatment initiation and then annually over 36 months. Patients with no evidence of disease activity showed persistently low NFl values in this study. Another relevant study was that of Thebault et al. [136] who used baseline and an average of six serial samples over the course of 48 weeks to assess the ability of either baseline sNFl or a change in sNFl to predict a relapse in a cohort of MS patients with highly active disease. They found that both baseline and longitudinal change in sNfL “may help identify patients who would benefit from early treatment optimization”. Some complexities in interpretation of NFl in peripheral blood have been reviewed in an excellent review article by Gafson et al., in 2020 [137]. Thus, many investigators currently feel that the best role of sNfL for individual patients in clinical practice may be as a serial monitoring tool for disease activity that is subclinical(e.g., MRI lesion or atrophy development) [138]. However, obviously more uses may become evident as the extensive research on this promising biomarker continues.

If further studies corroborate its role as a serial monitoring tool, sNFl could then supplement clinical measures such as history, examination (including the 25 foot timed walk [139]) and MRI. An increase in a patient’s baseline sNFl, even in the stability of other measures, might prompt more frequent monitoring or change in treatment or both. Of course, widespread use of sNFl would be helped by studies demonstrating its utility in individual patients. Unfortunately, given that sNFl is a measure of CNS injury, such a study might be difficult to run since the duration of the study might need to be significantly longer than the standard 2 years of many clinical trials in the field, especially if the clinical endpoint was disability progression. However, as more data is obtained on this promising biomarker, multiple helpful clinical trials are likely to be designed.

Thus, I anticipate a brave new world where the CXCL13 index can be used to guide early therapy, using more aggressive, but higher risk therapies for index positive patients and less aggressive, lower risk management in index negative patients. Then, sNFl can be used to serially monitor CNS injury and adjustments made to management should sNFl increase over time. Unlike the dystopian nature of Huxley’s “Brave New World”, the use of these biomarkers is likely to lead to a positive outcome: the much improved personalized management of MS.

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
