# Peer review of "The Brave New World of Early Treatment of Multiple Sclerosis: Using the Molecular Biomarkers CXCL13 and Neurofilament Light to Optimize Immunotherapy"

_biomedicines, 2022, doi:10.3390/biomedicines10092099_

Round 1
Reviewer 1 Report
Authors correctly discuss the need for biomarkers in MS diagnosis and treatment and specifically point their attention to two biomarkers, CXCL13 and NFL. They extensively discuss the literature on these two biomarkers with updated reference. The only issue I have is that the title can be quite misleading, since it seems to present an extensive review on potential biomarkers in MS, but the work is instead more focused on CXCL13 and NFL. Similarly, reading the abstract the importance of these two biomarkers seems marginal. In my opinion, authors could partially change the title and/or the abstract in order to give more relevance to CXCL13/NFL, so the reader could better understand what is the main focus of the review.
Author Response
The reviewers’ comments are in standard type, and the author’s responses are in italics.
Reviewer 1- Authors correctly discuss the need for biomarkers in MS diagnosis and treatment and specifically point their attention to two biomarkers, CXCL13 and NFL. They extensively discuss the literature on these two biomarkers with updated reference. The only issue I have is that the title can be quite misleading, since it seems to present an extensive review on potential biomarkers in MS, but the work is instead more focused on CXCL13 and NFL. Similarly, reading the abstract the importance of these two biomarkers seems marginal. In my opinion, authors could partially change the title and/or the abstract in order to give more relevance to CXCL13/NFL, so the reader could better understand what is the main focus of the review.
Changes have been made to the title and abstract as recommended.
Reviewer 2 Report
An interesting review proposing a combination of biomarkers CXCL13 and NFL for predicting disease activity and monitoring treatment responses for MS. The author needs to present evidence, a table, maybe to argue the advantage of CXCL13 over MRI findings and OCBs, the gold standards for MS. The title seems ambitious. A flow chart/figure and a summary would make it easier for the readers. The method section for detection of CXCL13 can be shortened.
Minor points
Add space before "("
Line 29, "posit"?
Line 40, remove "." before "("
Line 89, change "we" to "I"
Line 108, sentence not clear
Line 119, remove "-" after biomarker
Line 127, spell out TFH
Lines 136-140/242-244, please make it clear.
Lines 162, 207, font difference
Line 259. Covid.
Line 288-295. Please make it clear
Line 325, spell out LP
Author Response
The reviewers’ comments are in standard type, and the author’s responses are in italics.
Reviewer 2- An interesting review proposing a combination of biomarkers CXCL13 and NFL for predicting disease activity and monitoring treatment responses for MS. The author needs to present evidence, a table, maybe to argue the advantage of CXCL13 over MRI findings and OCBs, the gold standards for MS.
This reviewer is correct, that ideally these two biomarkers would be compared to MRI and OCB. However, I don’t think changes need to be made in the review for the following reasons. First of all, the review stresses that these are molecular biomarker while MRI is an imaging biomarker, not a molecular biomarker. Second of all, the review does state, as in the sentence below, that a clinical trial comparing MRI and OCB to NFl and CXCL13 would be necessary before there is broad clinical use of the biomarker:” … a clinical trial would also need to show that the CXCL13 index outperforms MRI and OCBs, which appears likely since the latter biomarkers are unreliable for prediction of future MS activity 102. “ Third, considerable space in the introduction is devoted to the explanation of why OCBs are a helpful diagnostic, but not predictive biomarker, and I don’t feel more emphasis on this is required.
The title seems ambitious.
As noted above the title has been changed.
A flow chart/figure and a summary would make it easier for the readers.
I don’t believe a flow chart/figure is going to be helpful and the last paragraph is a summary, and has been so titled.
The method section for detection of CXCL13 can be shortened.
The methods section for detection of CXCL13 has been considerably shortened.
Minor points have been changed as needed except as below:
Line 127, spell out TFH.
TFH is spelled out when it is first used in line 126:”… including follicular T helper cells(TFH) 38 and likely other T cell populations…